# Two decades of neuroscience publication trends in Africa

M. B. Maina [1,2,3 ✉], U. Ahmad [4,5], H. A. Ibrahim [6], S. K. Hamidu[3,7], F. E. Nasr [3,8], A. T. Salihu[9,10], A. I. Abushouk [11,12], M. Abdurrazak [13], M. A. Awadelkareem [3,14,15,16], A. Amin[3,17,18], A. Imam [19,20], I. D. Akinrinade [3,17], A. H. Yakubu [3,21], I. A. Azeez [16,22], Y. G. Mohammed [3,7,23], A. A. Adamu [24], H. B. Ibrahim[25], A. M. Bukar[26], A. U. Yaro[27], B. W. Goni[28], L. L. Prieto-Godino [3,29 ✉] & T. Baden [1,3,30 ✉]

Neuroscience research in Africa remains sparse. Devising new policies to boost Africa's neuroscience landscape is imperative, but these must be based on accurate data on research outputs which is largely lacking. Such data must reflect the heterogeneity of research environments across the continent's 54 countries. Here, we analyse neuroscience publications affiliated with African institutions between 1996 and 2017. Of 12,326 PubMed indexed publications, 5,219 show clear evidence that the work was performed in Africa and led by African-based researchers - on average ~5 per country and year. From here, we extract information on journals and citations, funding, international coauthorships and techniques used. For reference, we also extract the same metrics from 220 randomly selected publications each from the UK, USA, Australia, Japan and Brazil. Our dataset provides insights into the current state of African neuroscience research in a global context.

[1] School of Life Sciences, University of Sussex, Brighton, UK. [2] Biomedical Science Research and Training Centre, College of Medical Sciences, Yobe State University, Damaturu, Nigeria. [3] TReND in Africa (www.TReNDinAfrica.org), Brighton, UK. [4] Medical Genetics Laboratory, Genetics and Regenerative Medicine Research Centre, Faculty of Medicine and Health Sciences, Universiti Putra Malaysia, 43400 UPM Serdang, Selangor, Malaysia. [5] Department of Anatomy, Faculty of Basic Medical Sciences, Bauchi State University, PMB 65 Gadau, Nigeria. [6] College of Medicine, Misr University for Science and Technology, Giza, Egypt. [7] Department of Human Anatomy, Faculty of Basic Medical Sciences, Gombe State University, Gombe, Nigeria. [8] Faculty of Science, Alexandria University, Alexandria, Egypt. [9] Non-invasive Brain Stimulation and Neuroplasticity Laboratory, Department of Physiotherapy, School of Primary and Allied Healthcare, Faculty of Medicine Nursing and Health Sciences, Monash University, Melbourne, Australia. [10] Department of Physiotherapy, Hasiya Bayero Paediatric Hospital, Kano, Nigeria. [11] Harvard Medical School, Harvard University, Boston, MA, USA. [12] Faculty of Medicine, Ain Shams University, Cairo, Egypt. [13] Sheka Primary Health Care Kumbotso, Kano, Nigeria. [14] Faculty of Medical Laboratory Sciences, Al-Neelain University, Khartoum, Sudan. [15] UK Dementia Research Institute and Department of Clinical Neurosciences, University of Cambridge, Cambridge, UK. [16] Department of Neuroscience Biomedicine and Movement Sciences, University of Verona, Verona, Italy. [17] Instituto Gulbenkian de Ciência, Oeiras, Portugal. [18] Department of Physiology, Faculty of Basic Medical Sciences, University of Ilorin, Ilorin, Nigeria. [19] Department of Anatomy, Faculty of Basic Medical Sciences, University of Ilorin, Ilorin, Nigeria. [20] School of Anatomical Sciences, Faculty of Health Sciences, University of the Witwatersrand, Johannesburg, South Africa. [21] Faculty of Pharmacy, University of Maiduguri, Maiduguri, Nigeria. [22] Department of Veterinary Anatomy, Faculty of Veterinary Medicine, University of Jos, Jos, Nigeria. [23] Department of Biology, Neurobiology group, University of Konstanz, Baden Wurttemberg, Germany. [24] Department of Physiotherapy, Aminu Kano Teaching Hospital, Kano, Nigeria. [25] Department of Pharmacy, Federal Medical Centre, Katsina, Nigeria. [26] Centre for Visual Computing, University of Bradford, Bradford, UK. [27] College of Medical Sciences, University of Maiduguri, Maiduguri, Nigeria. [28] Department of Medicine, Yobe State University Teaching Hospital Damaturu PMB 1072, Damaturu, Yobe State, Nigeria. [29] Francis Crick Institute, London, UK. [30] Institute of Ophthalmic Research, University of Tübingen, Tübingen, Germany. ✉email: m.bukar-maina@sussex.ac.uk; lucia.prietogodino@crick.ac.uk; t.baden@sussex.ac.uk

Africa accounts for 15% of the global population but 25% of the global disease burden[1]. Moreover, the continent has the world's largest human genetic diversity, with important implications for understanding human diseases[2], including neurological disorders[3,4]. However, even though early progress in neuroscience began in Egypt[5], Africa's research capacity in this area has not kept pace with the developments in the field[6]. The reasons for this are diverse, and include low funding[1], inadequate research infrastructure[7], the relatively small number of active scientists[8], and their high level of administrative and teaching load[[9,10]. These barriers limit research and innovations from Africa[11], and contribute to the 'brain drain' from the region that extends long beyond neuroscience itself[12].

Over recent decades, an increasing number of local and international initiatives were set up to address some of these challenges, including in neuroscience[13]. This seems to have led to some progress, as for example, seen in a steady rise in the number of publications affiliated to some of the continent's countries with traditionally high numbers of neuroscience publications such as South Africa, Egypt or Nigeria[14]. Of the overall publications from Sub-Saharan Africa, almost 70% have non-African-based authors[6]. While on the one hand this may be indicative of important collaborative links between Africa and the rest of the world, it leaves it unclear which studies were truly African-led, and carried out in African labs—and which were rather led by researchers based elsewhere[6,15]. Indeed, a previous estimate suggested that as much as 80% of published health research that included African authors in Burkina Faso was not African led[16]. Here, we use the term (African led) to mean publications with clear evidence (e.g based on lead and senior author affiliations, and/or on manually asserted study location, where available) that the bulk of the intellectual input and experimental work was carried out by researchers who are primarily based at African institutions ('Methods').

Scientific publication database mining approaches using a combination of search terms such as 'Neuroscience' and 'Africa' have been used to estimate neuroscience research outputs from Africa by way of quantifying publication trends[14,17]. However, this approach does not delineate African-led studies from those led by researchers elsewhere. For example, PubMed data mining identifies 1247 Nigerian-led neuroscience papers between 1996 and 2017. However, manual curation revealed that of those, 54% were led by non-Nigerian laboratories[15]. Many of the remaining 46% Nigerian-led studies were published in Africa-based journals, some of which attract few citations from institutions outside of Africa[15]. Despite the importance of genetically modified model systems in driving neuroscience research breakthroughs, Nigerian-led studies were characterised by a general absence of genetically modified animal model systems and only occasional use of more resource-intensive techniques such as advanced fluorescence microscopy or neuroimaging[15]. However, while these country-specific analyses are valuable, in view of the continent's geographical, political and cultural diversity, further research is needed beyond a single country.

To survey African-led neuroscience publications as a whole, we here developed a measure that involves manual curation of PubMed-retrieved articles to ascertain whether they were led by researchers who were primarily affiliated and based at African institutions. Accordingly, we manually went through each of 12,326 PubMed-listed neuroscience publications affiliated with African institutions between 1996 and 2017. We identified those that presented clear evidence that the research was indeed carried out in Africa (see above, and 'Methods'). This eliminated ~58% of publications to leave 5219, on average, five per country and year. From here, we extracted key metrics, including author affiliations, the field of neuroscience, journals and citations, as well as information on funding, models and techniques. For comparison, we also extracted the same metrics from 220 randomly selected publications, each from the UK, USA, Australia, Japan and Brazil, of which 79% passed our inclusion criteria (Fig. S1).

We here present a summary of our main findings. Specifically, we asked: How many publications came out of each African country, and how were different sub-disciplines represented (Fig. 1)?, in which journals are they published and how many citations did they attract (Fig. 2)?, what were the major trends of international co-authorships (Fig. 3)?, how was the work funded (Fig. 4)?, and what experimental techniques were used (Fig. 5)?

## Results

**African neuroscience by numbers of publications.** Africa's neuroscience publications since 1996 ($n = 5219$ publications, see introduction) have been dominated by a small number of countries: Egypt ($n = 1478$, 28%), South Africa ($n = 1181$, 23%), Nigeria ($n = 566$, 11%), Morocco ($n = 409$, 8%) and Tunisia ($n = 388$, 7%) (Fig. 1a). Together, these five countries account for more than three in every four neuroscience papers published from the continent. At 2–3% each, further contributions came from the East African nations of Kenya ($n = 131$), Ethiopia ($n = 119$) and Tanzania ($n = 103$), followed by 1–2% each from Cameroon ($n = 81$), Malawi ($n = 71$), Algeria and Senegal ($n = 70$ each), Uganda ($n = 69$) and Ghana ($n = 60$). Beyond these, numbers per country are lower, with more than half of African countries contributing fewer than 10 papers. Nevertheless, over the past two decades, the number of neuroscience publications published each year has exponentially increased across all of Africa's major geopolitical regions (Fig. 1b). Accordingly, the continent's number of neuroscience publications is on an all-time high with a clear upwards trajectory (see also Fig. S2A).

Here, dominant research schemes[18] include neurodegeneration and injury ($n = 2066$, 34%; compared to 22% outside of Africa (OA)), followed by techniques ($n = 905$, 15%; OA: 16%), excitability, synapses and glia ($n = 550$, 9%, OA: 15%), development ($n = 532$, 9%; OA: 16%), and physiology and behaviour ($n = 511$, 8%; OA: 13%) (Fig. 1c). In comparison, research on motivation and emotion ($n = 217$, 4%; OA: 3%), motor systems ($n = 191$, 3%; OA: 9%), cognition ($n = 155$, 3%; OA: 4%) and sensory systems ($n = 92$, 2%; OA: 2%) is less prevalent. By and large, and despite a small degree of inevitable variation, this general distribution across major neuroscience research schemes has been surprisingly constant, both across countries (Fig. 1a), and over time (Fig. S2B).

**The visibility of African neuroscience publications.** There is no single, universally useful metric for measuring research visibility and influence. Metrics like journal impact factor (IF, i.e. the yearly average number of citations of articles published in the last two years in a given journal) and citation counts have many shortcomings and vary across research fields. Nonetheless, in the absence of perhaps less problematic numerical approaches, we compared these two metrics in relation to African neuroscience publications across countries. For global context, we computed the same metrics for 220 randomly selected papers each from the USA, UK, Japan, Australia and Brazil. This revealed a great diversity of African research visibility, with many papers ranking on par with many non-African papers (Fig. 2a). However, different regions varied markedly in the distribution of these metrics. For example, with a mean of ~13 citations per paper, West-African publications tended to be cited least frequently. In contrast, Southern Africa's publications were on average cited 31 times, on par with those coming from Brazil. Nevertheless, though dominated by the Global North (here: UK, USA, Japan, Australia, mean of ~77 citations per paper), also researchers from most African regions published at least a small fraction of papers

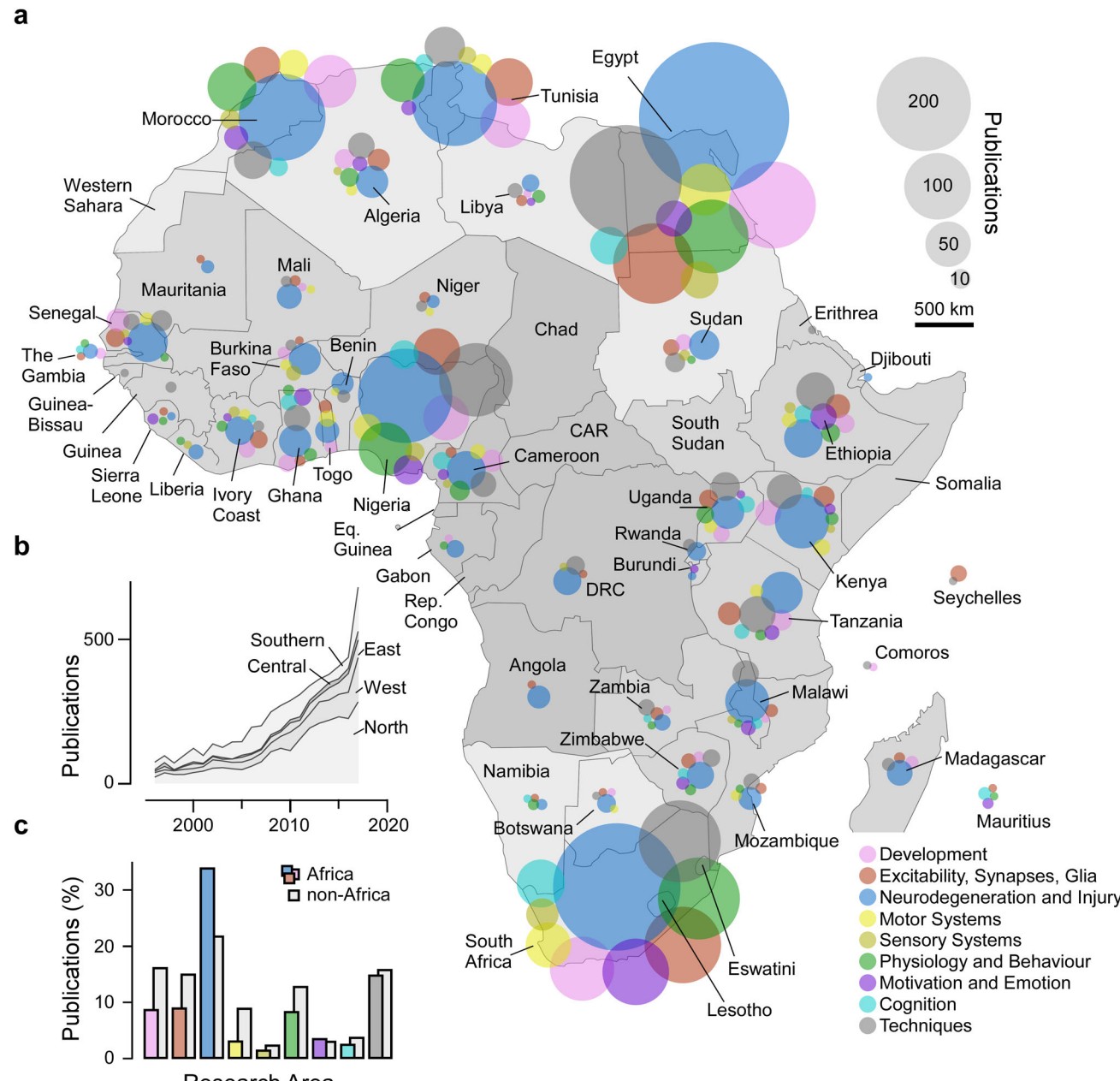

**Fig. 1 African neuroscience publications 1996–2017. a** Overview of Africa's neuroscience publications in the timeframe indicated, organised in nine broad topics as indicated by the different colours. Bubble sizes denote the total number of papers per country and topic, as indicated. **b** Total publications per year, with contributions from different African regions highlighted. Regions were delineated following the United Nations definition into North Africa, West Africa, East Africa, Central Africa and Southern Africa. See also background shading in (**a**). **c** Distribution of research topics in Africa (coloured bars, for legend see (**a**)) and outside of Africa (grey bars). Source data are provided as a Source Data file.

in the top bracket (citations ≥ 95). These trends were largely mirrored also in the publishing journal's IF (Fig. 2b, c). However, unlike for citations, very few publications from African labs ranked in the top bracket (here arbitrarily defined as IF ≥ 9.5), which supports frequently discussed shortcomings of this metric for the assessment of individual manuscripts[19,20]. Together, even though for now much of global neuroscience research remains dominated by the Global North, the number of neuroscience publications from African labs is undeniably growing.

**Co-authorship with international authors.** One key aspect of integration into the global research community comes through

international scientific collaborations. Here, the lack of funding and barriers related to visa processes have long made it difficult for many African researchers to engage with colleagues abroad[21]. However, where these difficulties have been overcome, publication and citation metrics do stand to gain. For example, African-led neuroscience publications with international-based co-authors—both within Africa and beyond—tended to be cited more frequently, and were published in higher IF journals (Fig. 3A). Perhaps unsurprisingly, the capacity for international co-authorship is therefore associated with research visibility[22]. We therefore next investigated how Africa's co-authorship networks are geographically organised.

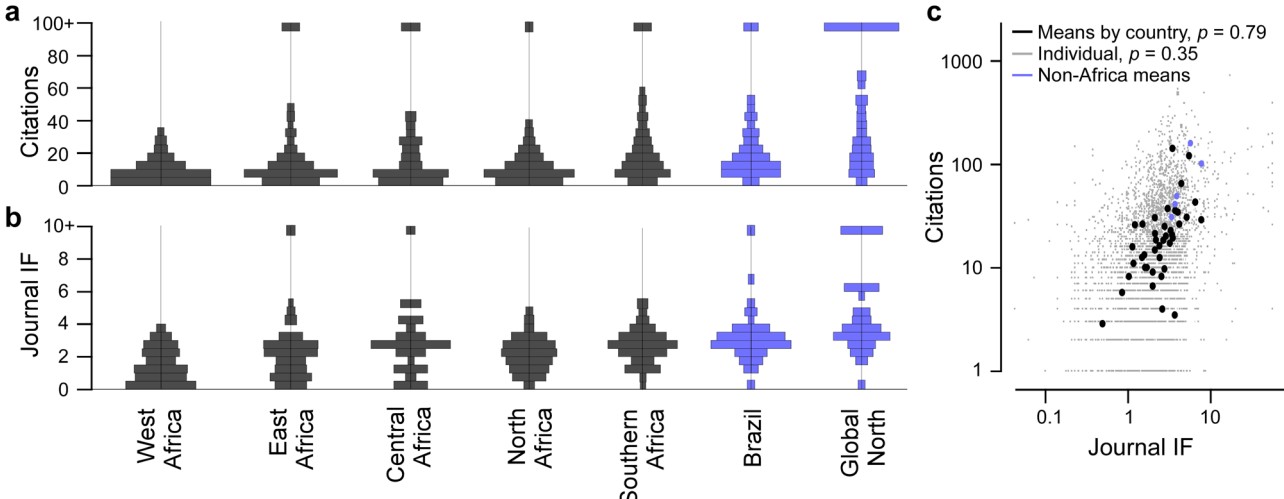

**Fig. 2 Citations and journal impact factors. a, b** Area-normalised histograms of citations (**a**) and the publishing journal's impact factor (IF, **b**) of all papers from different African and non-African regions, as indicated. In each case, all citations above ≥95, and IF above ≥9.5 were allocated to a single bin (top). **c** Citations plotted against IF for every paper in the database (small grey dots), and for means by country (large dots), as indicated. Linear correlation coefficients as indicated. Source data are provided as a Source Data file.

This revealed that besides collaborating domestically, African Neuroscientists rarely published with co-authors across long distances *within* Africa, instead, co-authorship with researchers outside the continent of Africa was more common (Fig. 3B–D). In particular, many internationally co-authored papers had links with either Europe or North America. Similarly, any intra-African co-authorships were mostly with contributions from other African regions with overall higher publication outputs. For example, West, East and Central African-led studies occasionally included co-authors from Southern Africa, while vice versa Southern African-led papers almost never had East- West- or Central African co-authors (Fig. 3B). All the while, papers from North Africa mostly included co-authors from Europe, North America and the Middle East. The latter examples follow a geographic and cultural pattern[23,24], for example, in terms of shared dominant religions and languages. Generally, fractions of international co-authorship of publications increased over the study period, with approximately preserved geographical patterns (Fig. S2C).

Overall, the striking preponderance of co-authorship beyond Africa's borders over pan-African international links is reminiscent of the continent's logistical networks—here exemplified by available international flights (Fig. 3E, F). It seems likely that both networks are linked to common underlying factors such as historical, linguistic and cultural ties as well as economic considerations[24]. Nevertheless, this currently poor international connectedness within the continent ought to be considered in future efforts aiming to build a more united African research landscape.

**Funding African neuroscience**. We next asked how Africa's neuroscience research is funded. To this end, we assessed funding declarations in Africa's 265 top papers ('Methods'). Of these, many ($n = 93$, 35%) declared no funding at all (Fig. 4a). This lack of declarations was pervasive throughout the continent, but particularly prominent in Northern Africa ($n = 46$ of 73, 63%). Many of these studies may have been self-funded, and/or published in outlets that do not require a funding declaration.

Of papers that did declare the funding ($n = 172$), we next assessed whether the sources were domestic and/or international (Fig. 4b). This revealed that most of these African neuroscience publications were supported by international rather than domestic agencies. For example, only 3 out of 37 (8%) of East

African top papers declared domestic funding, while 36 (97%) declared international funding. The only African region where the number of domestic funding mentions exceeded international funding mentions was Southern Africa: ($n = 49$ (73%) domestic and $n = 32$ (48%) international; of $n = 67$ total). In comparison, between 92% (UK) and 100% (Brazil) of papers included in our analysis declared domestic funding, with between 9% (Japan) and 57% (Australia) of papers declaring additional international support. It seems clear that the availability of local, rather than (or in addition to) international funding is critical to building a viable research culture, and Southern Africa appears to be the only region that is beginning to reflect this need. Indeed, South Africa, by far Southern Africa's largest research contributor, is the only country in Africa that invests nearly 1%, of its GDP in research and development, as recommended by the African Union in 2007[1].

Nevertheless, 46% ($n = 123$) of Africa's 265 top neuroscience papers declare international funding, hinting that these factors may be linked. The vast majority of this support came from the USA, who supported $n = 44$ (36%) of these 123 papers, followed first by the UK ($n = 31$, 25%), and then France ($n = 11$, 9%), Switzerland ($n = 9$, 7%) and Germany ($n = 7$, 6%) (Fig. 4c, d). Accordingly, unlike manuscripts with international co-authorship (Fig. 3D), international funding support from the Middle East, Asia, Australia and South America for African neuroscience was limited. By agency, the USA's NIH was acknowledged most frequently ($n = 42$, 34%), followed by the UK's Wellcome Trust ($n = 24$, 20%) and Medical Research Council ($n = 10$, 8%). Next was the World Health Organisation ($n = 6$, 5%), and beyond this, no agency received more than 2% of international funding mentions.

**Model systems, techniques and medicinal plants**. Advances in our understanding of nervous systems are notably driven by equally rapid advances in (bio)technology. Accordingly, access to state-of-the-art research tools—both technological and biological—remain central to scientific success. Accordingly, understanding the availability and use of such tools across Africa is likely to be pivotal to any strategy to support future research. To this end, we categorised methods employed in each of the surveyed >6000 papers as either 'type 1' and 'type 2', where type 1 techniques can generally be supported already with minimal infrastructural investment (e.g. classical histology, chromatography and/or behaviour), type 2 was

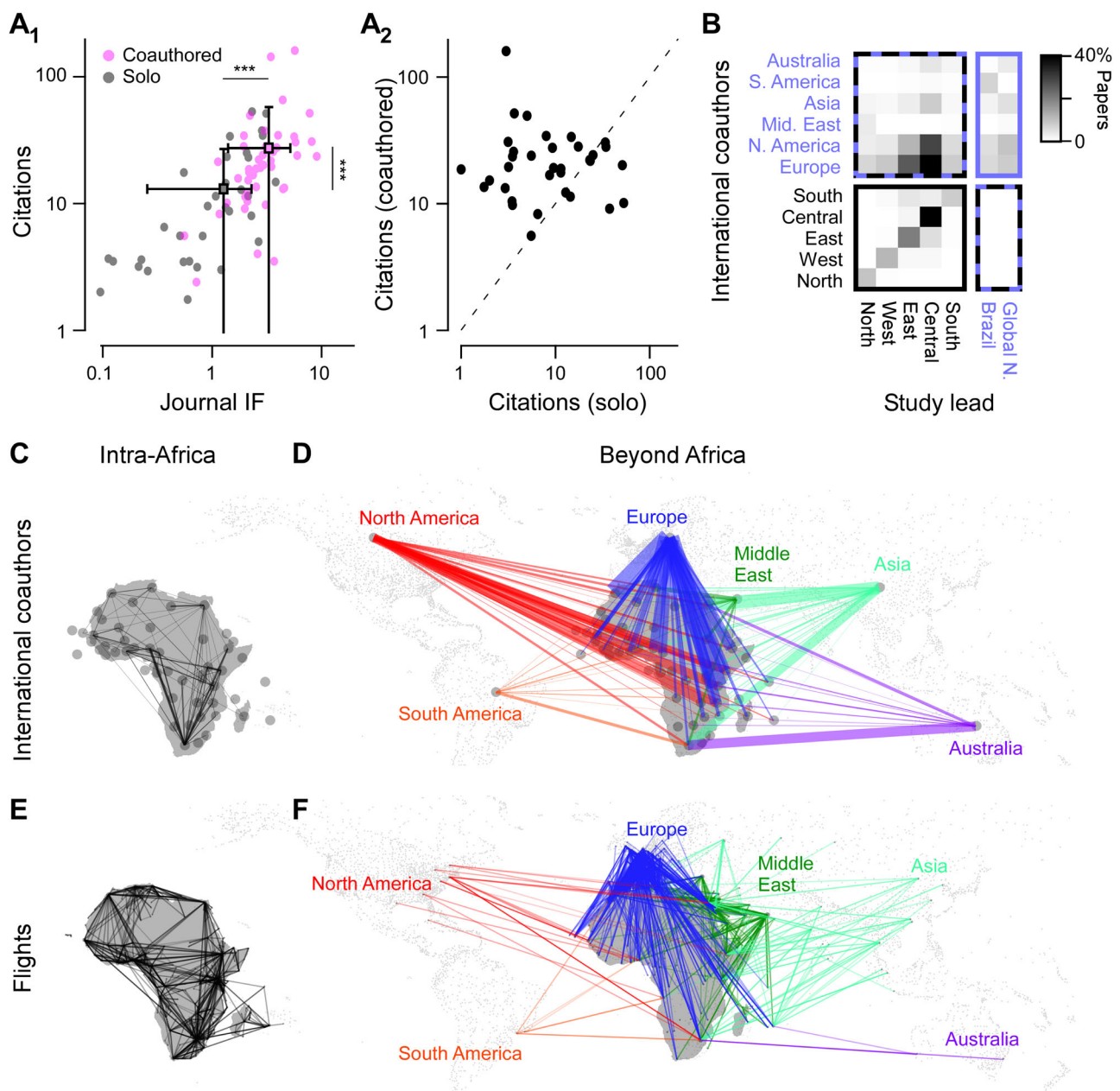

**Fig. 3 Co-authorships.** **A₁** Citations plotted against the journal impact factor (IF) for each African country, divided into publications without (grey, 'solo') and with international co-authors (pink, 'coauthored.'). The latter included both within-Africa international co-authors as well as co-authorships beyond Africa's borders. Square-markers and errors denote each population's mean and s.d. Both citations and IFs were significantly higher for internationally co-authored papers (Wilcoxon Rank Sum, 1-tailed, ***$p < 0.001$, full $p$ values: $7.7 \times 10^{-15}$ and $1.1 \times 10^{-12}$ for citations and IF, respectively; $n = 45$ countries in both cases). **A₂** The same citation data as (**A₁**) plotted pairwise for with (y-axis) and without international co-authors (x-axis), highlighting the substantial positive influence of international co-authors. Dotted line indicates parity. **B** Co-authorship matrix between African regions and the rest of the world, with darker colours indicating a higher preponderance of international co-authorships. **C**, **D** Intra-African (**C**) and Beyond-African (**D**) co-author links organised by African country and major geopolitical regions beyond Africa, as indicated. The thickness of lines illustrates the total number of internationally co-authored papers, while colourings in (**D**) illustrate the co-authorships with partner beyond Africa. **E**, **F** as (**C**, **D**), respectively, but for the existence of international flight routes based on data from OpenFlights.org. Each route is illustrated with a single line of consistent opacity and thickness. Source data are provided as a Source Data file.

aimed to summarise the use of more resource-intensive techniques that traditionally come in hand with substantial investment in research infrastructure (e.g. fluorescence microscopy, molecular biology, cell culture work, neuroimaging; for a full list of criteria, see 'Methods'). Despite this set of arguably conservative criteria, and

with the notable exception of The Gambia ($n = 5/14$; 36%, all linked to an MRC-funded research unit), no African country's neuroscience publications comprised more than a quarter of type 2 entries (Fig. 5a₁,₂). African countries with highest use of type 2 techniques were Egypt ($n = 363/1478$; 25%), South Africa

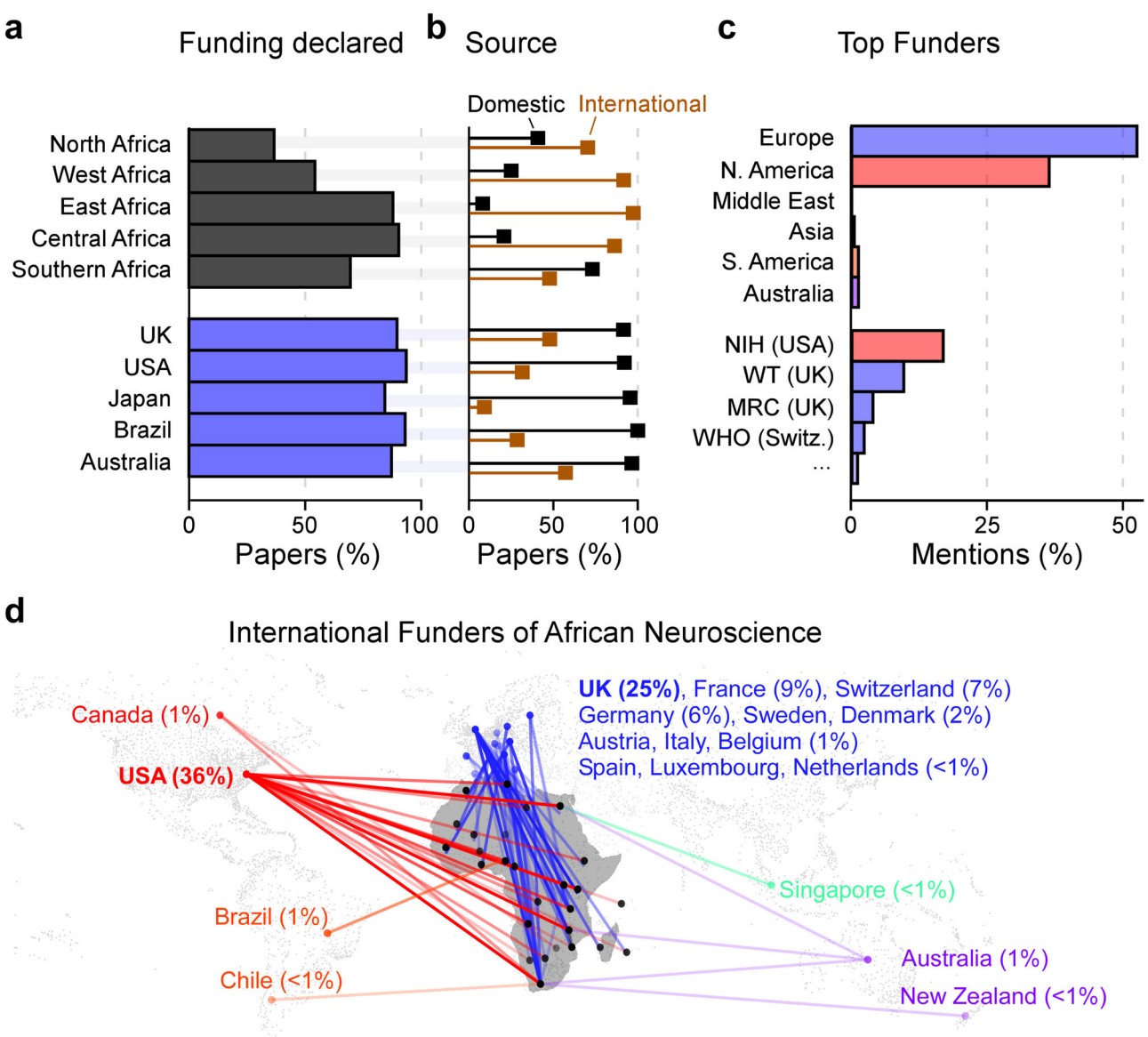

**Fig. 4 Funding. a** Percentage of papers with IF ≥ 5 that included any form of funding acknowledgements. **b** From (**a**), where declared, the source of funding, classified as domestic (black) and international (brown). International funding includes those received from any other country, including other African countries. Percentages are computed from each paper declaring either domestic or international support, or both. **c** Percentages of funder mentions of all African papers with IF ≥ 5 where international funding was declared (corresponding to the sum of brown columns in (**b**)). Where present, multiple funder mentions per single paper were individually included. **d** International funding links from (**b**, **c**) displayed by geography. Each funding mention is illustrated with a single line of consistent opacity and thickness. Source data are provided as a Source Data file.

(n = 272/1181; 23%), Morocco (n = 68/409; 17%), Tunisia (38/388; 10%), Nigeria (45/566; 8%), Ethiopia (9/119; 8%) and Algeria (2/35; 6%). All other African countries ranked below 5%, including many at 0% (countries with fewer than 10 papers were excluded from this analysis). In contrast, Japan, UK and USA all published 75% of papers based on type 2 techniques, followed by Australia (54%) and Brazil (33%).

Next, there was a near-complete absence of small, low cost and genetically tractable model systems such as fruit flies, zebrafish or *Caenorhabditis elegans*[25] in African neuroscience publications (Fig. 5b$_{1,2}$). Unlike USA (33%), UK and Japan (23%), Australia (12%) or Brazil (3%), no African country used any genetically

modified model systems (including cell culture or mammals) in more than 1% of neuroscience publications. Most countries used none at all. Clearly, the promotion of the use of such model systems should be considered as part of strategies aimed to modernise Africa's research landscape.

Finally, we assessed the use of endemic medicinal plants in African neuroscience publications, many of which have been used for centuries for the treatment of diseases. Research in natural medicinal products puts Africa in an excellent position in the area of drug discovery[26]. This revealed that research in this field is highly diverse across the continent (Fig. 5c$_{1,2}$). In particular, several West-African countries with tropical and subtropical climates have

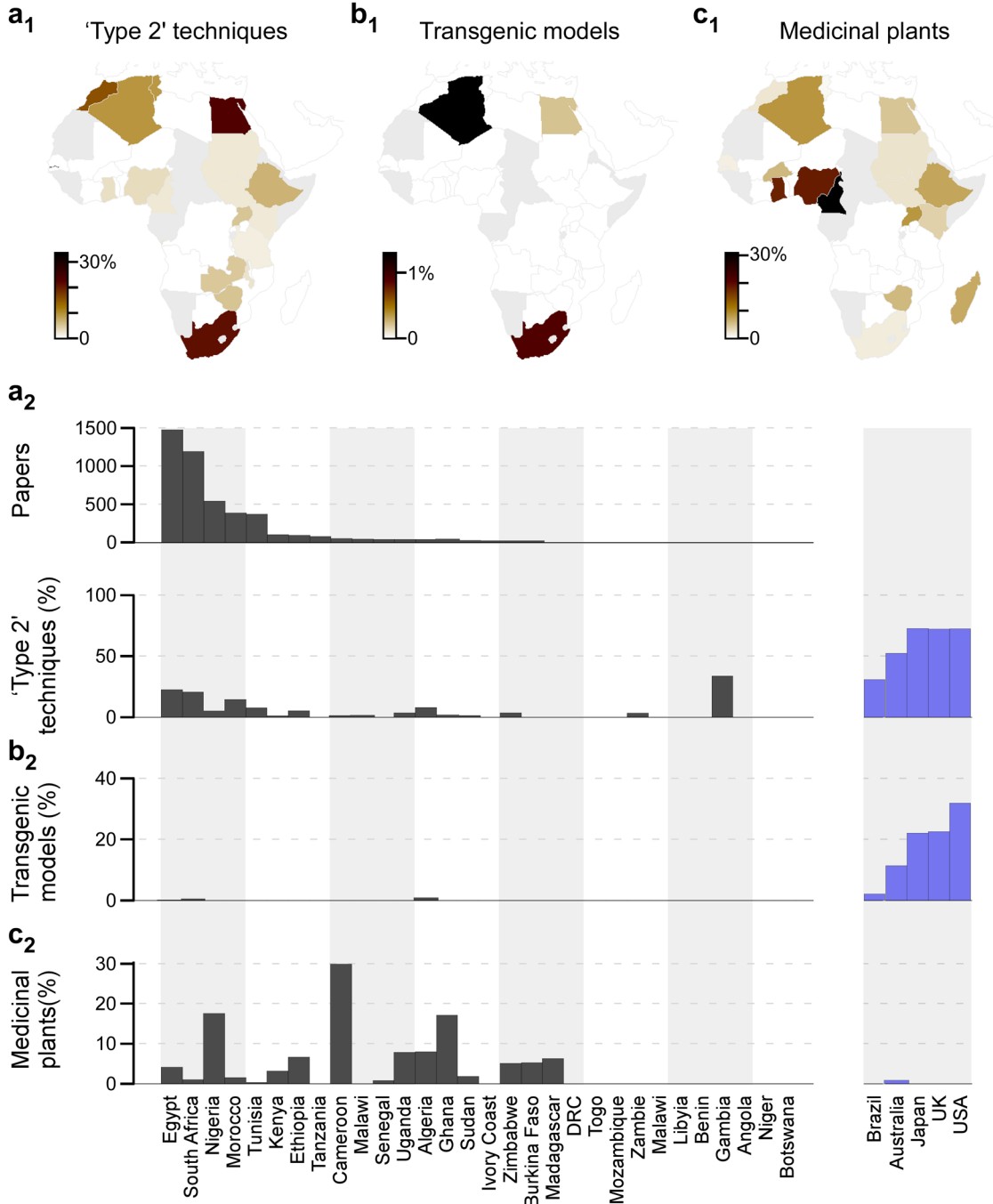

**Fig. 5 Research techniques. a₁–c₁,** Percentages of papers that used 'type 2' techniques ('Methods') (**a₁**), transgenic models (**b₁**) or medicinal plants (**c₁**), organised by geography. Countries with fewer than 10 papers in the dataset were excluded (grey). **a₂–c₂** As (**a₁–c₁**), plotted as percentage bars per country, with the same metrics extracted from representative non-African publications (blue). African countries are sorted by the total number of papers published (**a₂**, top), excluding countries with fewer than 10 papers in the database. Source data are provided as a Source Data file.

invested heavily into this branch of neuroscience, most notably Cameroon (30%) as well as Nigeria and Ghana (18% each). In contrast, many other countries, including both of Africa's most prolific contributors to neuroscience publications (Egypt 5% and South Africa 1%), are more focused on other topics.

## Discussion
Our dataset highlights that Africa's neuroscience publication numbers and their citations are on an all-time high, with a clear and ongoing upwards trajectory. Similarly, while the number of

neuroscientists in the continent remains tiny compared to the total population (e.g. Refs. [27–29]), the neuroscience scientific workforce is on the rise. This is, for example, mirrored in the increasing number of neuroscientists attending the Society of Neuroscientists of Africa (SONA) bi-annual meetings[30], or a continuous rise in the number of applications for African-based neuroscience training programmes. However, to continue supporting this growth, major and ongoing investment into African science must be ensured.

Most declared neuroscience funding came from external sources, most notably from the USA and the UK. However, local

funding is instead needed for establishing a sustainable African neuroscience research environment. At the moment, none of Africa's 54 countries invests as much as 1% of their GDP into R&D, as recommended by the African Union[1]. South Africa is one of the few countries nearly meeting this target, which likely part-explains its leading role in African neuroscience. Compared to other African countries, the high proportion of domestic funding in South Africa might be part-explained by its national policy changes in the research and development budget. In 2003, the South African government issued a new funding scheme that directs funding to tertiary institutions based on their research output. Since 2012, South Africa's gross expenditure on research and development (GERD) has consistently increased according to the national R&D survey[31].

For African research to continue to grow, more government funding is needed to provide reliable support to their domestic research sector. Such efforts might also be well-supported by the local philanthropic sector. Africa has a large number of individuals and charitable organisations with access to substantial funds[32]. Governments, scientists and the general population must engage with these to contribute to local science funding, much like major non-African philanthropic organisations such as the Gates Foundation or Wellcome Trust that currently fund research on the African continent. African Neuroscientists may also seek to facilitate further international networking and collaboration opportunities, particularly within Africa, in view of attracting multinational funding[33]. In addition, increased engagement with science advocacy campaigns to raise the profile of African research and its relevance to both global and local problems may be expected to further facilitate their cause. This is particularly relevant given the continents genetic diversity which can help in understanding global health problems[2]. Advocacy campaigns could focus particularly recognisable disease aspects such as central nervous system (CNS) infections (e.g. cerebral meningitis) or konzo. Increased awareness of African-led research in fields such as these, and their benefits to society, may translate to increased support and provision for funding.

Indeed, already now, many African neuroscience articles under the Neurodenegerative Disorders and Injury theme were studies on meningitis, konzo, stroke, neurological manifestations of HIV, and epilepsy. In general, these areas were the most intensely studied neuroscience research themes in Africa, which may also reflect an increased awareness about the prevalence of these conditions amongst scientists, policymakers and the general public. In support, previous work reported a dementia incidence rate in Africa between <1% and 10% in population-based studies, and up to 47.1% in hospital-based studies[34]. For traumatic brain injury (TBI), the 2016 Global Burden of Disease study[35] estimated nearly six million cases across all of Africa, the highest of any continent. By 2050, the prevalence of TBIs in Africa is predicted to further rise, to between 6 and 14 M cases annually[36]. Similarly, compared to most non-African countries, Africa's incidence rate of epilepsy is expected to double[37] alongside a projected increase in the prevalence of stroke[38]. Beyond the critical need for global recognition and action on these issues, they also provide a powerful platform for lobbying more generally for additional support of African neuroscience research.

Although there is clear evidence of increasing numbers of neuroscience publications from African laboratories, there is still much room for growth. Based on citation and IF metrics, there remains substantial heterogeneity in the visibility of the neuroscience publications across the continent. Under the caveat that using such metrics as a shorthand for 'research quality' is not appropriate[19,20], West Africa seems to lag behind among all the regions. For example, Nigeria, the region's country with the greatest number of publications, published only one neuroscience paper in an IF ≥ 9.5 journal in the 21-year period[15]. The lack of visibility, especially in citations, may be part-explained by choices over where work is submitted for publication. Many Nigerian neuroscience papers are published in African journals, many of which are rarely read beyond the continent's borders[15]. Moreover, many African journals are not PubMed indexed (and therefore excluded from our study)[39]. Given the clear benefit of publishing in indexed journals for driving research and collaboration, this flags the need for African academics to increasingly target indexed journals. This will be facilitated by increasing the widespread availability of both author and of access fee-waivers from international outlets[40].

Next, our analysis highlights a profound lack of the availability and/or use of state-of-the-art equipment and modern experimental approaches. With few notable exceptions, tools like fluorescence microscopy, molecular biology or cell culture were used in less than 10% of most African countries' neuroscience publications (see also Ref. [15]). Next, while some public institutions have some type 2 equipment located in individual departments, access is often restricted to a small number of researchers which can limit their widespread use[15,40]. Although funding schemes and training programmes have enabled many African scientists to acquire diverse neuroscience skills in foreign labs, the absence of the same research infrastructure back at their home institutions continues to restrict the extent to which such skills can be put into use. Clearly, beyond financial investment, African researchers must be afforded widespread access to diverse research infrastructure[9]. In addition to the provision of training opportunities abroad, local and international neuroscience funding initiatives should support African scientists to establish their laboratories. Similarly, African labs have much to gain from investing in infrastructure and expertise in designing and producing research-grade open hardware equipment[9,41–43].

Finally, the near-complete absence in the use of transgenic models in African neuroscience publications is worrying, and may contribute to the generally low citation number of African neuroscience publications. Instead, many African neuroscientists continue to rely on wild-type rodent models, most notably rats, followed by mice[15]. The cheap and genetically amenable nature of model systems like zebrafish, fruit flies or *C. elegans*, makes these models ideal. One-third of the Nobel Prizes in Physiology and Medicine awarded between 1996 and 2017 relied heavily on non-mammalian yet genetically accessible model systems[44]. The many challenges faced by African neuroscience, most notably lack of funding, make ultra-low-cost models like fruit flies and *C. elegans* particularly interesting for research on the continent[45]. This particularly calls for increased investment to facilitate the use of these and other similar affordable and genetically amenable model species in African neuroscience. For this, scientists and funding agencies will also need to work closely with national governments and biosafety authorities to put regulation for the import and use of genetically modifiable animals in place, which to date is missing in many African countries.

Taken together, while the number of African neuroscience publications remains comparatively small, it is clearly on the rise. To sustain this rise and increase the continent's neuroscience visibility, there is a clear need for increased investment in modern research equipment, training in the use of this equipment and the adoption of genetically tractable models. While some of this investment will likely continue to come from beyond Africa's borders, it will be critical to bolster African countries' domestic research support streams, from governments and private funders alike. Next, while international collaborations are valuable, African neuroscience must in parallel be strengthened through intra-African collaborations and the promotion of sharing of restricted resources.

In view of the highly interdisciplinary nature of neuroscience, many aspects of our findings may potentially generalise to other scientific disciplines.

## Methods

**Data extraction**. Neuroscience-related research articles from Africa, USA, UK, Australia, Japan or Brazil from January 1996 to December 2017 were retrieved from PubMed. Search terms used were 'neuroscience' OR 'nervous system' OR 'brain' OR 'neuron' OR 'spinal cord', in combination with the name of each of the individual African and non-African countries were used. The searches also included author affiliation fields, as well as the full text—from here, a small number of false-positives (e.g. where a paper mentioned a specific African country in the full text without the research having been carried out elsewhere) were excluded by hand. Primary research, case reports or clinical trials were included, while review articles were excluded. Next, duplicates (~10%) and irrelevant articles were manually removed. This yielded a total of 12,326 candidate papers from Africa. For comparison, 220 papers each from the above-listed non-African countries were also analysed, after randomly selecting 10 publications per year and country using the same search terms (Fig. S1). Of the total of 1100, $n = 229$ (21%) were eliminated based on the same exclusion criteria applied to our African dataset to leave a total of $n = 871$ non-African papers (Australia: 164; Brazil: 173; Japan: 197; UK: 171; USA: 166).

**Data curation**. Most data curation was done by hand, as detailed below. All raters were trained neuroscientists, with experience ranging from MSc level to faculty. Rating practice was aligned first by the lead author training all raters individually. Subsequently, each paper was independently rated twice, by two separate members of the team, followed by manual checking and adjusting by the first author for any inconsistencies between each paper's two ratings. Curators per country were chosen such that all of a given country's publications were curated by the same two curators (and, if needed, the lead author, as noted above). To identify research conducted within each country, the full texts of all the articles were retrieved and screened manually. For exclusion, papers from outside of Africa were identified based on the listed affiliations of lead/corresponding/senior author(s) as well as study location. The latter was extracted from information in the materials and methods or acknowledgements, where possible. For example, articles with external co-authorships in which only a small fraction of the work was conducted within Africa, such as sample collection, were excluded. Moreover, even if the authors did not have an affiliation with an African institution, their work was included as long as it was conducted within Africa (as judged, e.g. from methods and/or acknowledgements sections). This process eliminated $n = 7107$ papers, leaving $n = 5219$ African papers for further analysis (Fig. S1). For simplicity, we did not attempt to quantify our dataset by individual contributing author. This is because of the difficulty in reliably linking individual authors across publications that may use a variety of name-formatting in the author list, and because unique author identifiers such as ORCIDs were not reliably listed in all surveyed papers.

The latter were further screened by hand to retrieve the total number of google scholar citations, the publishing journal and its Clarivate Analytics impact factor, as well as information on model species whether or not they used medicinal plants. Impact factor for journals not indexed by Clarivate Analytics was estimated from Scimago. In addition, author affiliations were screened for co-authorships between research institutes, both nationally and internationally. In addition, we summarised each paper's research techniques as either 'type 1' or 'type 2': Type 1 techniques included histology, biochemical assays, such as enzyme-linked immunosorbent assay (ELISA), plant extract preparation, high-performance liquid chromatography (HPLC), behaviour and blood analysis. Type 2 included electron microscopy, western blotting, immunohistochemistry, cell culture techniques, cloning, flow cytometry, fluorescence microscopy, whole-brain imaging, advanced neuroimaging (e.g. fMRI), sequencing and identifying genes of interest, molecular cloning and recombinant DNA technology, gene delivery strategies, making and using transgenic organisms, manipulating endogenous genes, as well as any additional technique that was judged to be similarly resource-demanding, where required. A paper was classified as type 2 if it used any type 2 technique, even if it mainly used type 1 techniques. In addition, each paper was attributed to one of the nine broad topical neuroscience themes, as put forward based on attendance at the Society for Neuroscience annual meeting (see also Ref. [18]). Specifically, topics included (i) techniques, (ii) cognition, (iii) motivation and emotion, (iv) physiology and behaviour, (v) motor systems, (vi) sensory systems, (vii) neurodegeneration and injury, (viii) excitability, synapses and glia and (ix) development. Finally, for all $n = 265$ African and $n = 232$ non-African papers that were published in journals with an IF ≥ 5, we also extracted information on funding. We used only this subset of publications because this task was unusually time-consuming in view of incomplete standardisation across publishing outlets and the large diversity of funders worldwide. Data on international flights was taken from OpenFlights.org, an open-source database of all major flight routes worldwide. The corresponding dataset is publicly available at https://github.com/jpatokal/openflights/. Data and sources for other metrics, including GDP, R&D spending in purchasing power parity (PPP), GERD, population, etc. are detailed in the raw data tables provided on GitHub (see 'Data availability'). PPP is a measurement of prices in different countries that uses the prices of specific goods to compare the absolute purchasing power of the countries' currencies[46]. Starting from raw Microsoft Excel tables provided, all data analysis was performed in Igor Pro 6 (Wavemetrics) and GNU-R.

**Reporting summary**. Further information on research design is available in the Nature Research Reporting Summary linked to this article.

## Data availability

Data were retrieved from PubMed at https://pubmed.ncbi.nlm.nih.gov/. All data is freely available without restriction at https://github.com/BadenLab/AfricanNeuroscience and https://zenodo.org/badge/latestdoi/267297804 (see Ref. [47]). Source data are provided with this paper.

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

## Acknowledgements
Funding was provided by the European Research Council (ERC-StG 'NeuroVisEco' 677687 to T.B., ERC-StG 'EvolutioNeuroCircuit' 802531 to L.L.P.G.), The UKRI (BBSRC, BB/R014817/1 to T.B., and MRC, MC_PC_15071 to T.B.), the Leverhulme Trust (PLP-2017-005 to T.B.), the Lister Institute for Preventive Medicine (to T.B.) and the Wellcome Trust (Investigator Award in Science to T.B. 220277/Z20/Z). L.L.P.G.'s work was supported by the Francis Crick Institute which receives its core funding from Cancer Research UK (FC001594), the UK Medical Research Council (FC001594), and the Wellcome Trust (FC001594). We also wish to thank the FENS-KAVLI Network of Excellence for support (T.B., L.L.P.G.), as well the as EMBO YIP programme (T.B.).

## Author contributions
The study was conceived and organised by M.B.M. Data curation was done by M.B.M., A.U., I.H.A., H.S.K., N.F.E., S.A.T., A.A.I., A.M, A.M.A., A.A., I.Az., I.Ak., A.I.D., Y.A.H., A.I.A., Y.G.M., A.A.A., I.H.B., B.A.M., Y.A.U., B.W.G. and coordinated by M.B.M. Data analysis was done by T.B. The paper was written by M.B.M., L.L.P.G. and T.B.

## Competing interests
The authors declare no competing interests.
