## [Peer Review File. · Nature Communications]

REVIEWER COMMENTS

Reviewer #1 (Remarks to the Author):

In the current article, the authors analyzed all of Africa's Neuroscience research output over the past 21 years. Although there are few reports that were published on the same topic, yet the previous estimates addressed global African Neuroscience research, not the African-based neuroscientists. The current study focuses on research output that originally performed in Africa under supervision from African neuroscientists and not the internationally led collaboration. In fact, this effort is genuine and reflects the real metrics of neuroscience research activities in Africa. Furthermore, the authors compared their dataset to findings from well-developed countries in terms of demographic findings, the effect of the economy, and indicators of mobility. from my point of view, this work is valuable however there are some points to be discussed with the authors :

1-In the abstract line no. 4 the authors mentioned that "In contrast, with an estimated one neuroscientist per million people in Africa, news about neuroscience research from the Global South remains sparse". The accurate and updated Population, Estimated Neurologist Density in Africa is reported in WHO atlas of Neurology 2017 and updated 2020, please try to incorporate in your study

2- In the results section Fig(1). The authors highlighted the relation between neuroscience publications output and research and development(R&D) spending in purchasing power parity dollars (PPP\$). Could the authors explain the term (PPP\$)anywhere in the text as some readers are not oriented by this indicator

3- In the results section, the authors divided neuroscience research into nine broad topics according to reference no.18 and according to their research schemes neurodegeneration and injury (n =2,066, 34%; compared to 22% outside of Africa). The authors have to explain why in particular Neurodegenerative research is significantly more productive inside Africa than that lead by non-African

4- In terms of funding African neuroscience, the authors mentioned an important fact "only African region where the number of domestic funding mentions exceeded international funder mentions was Southern Africa (n = 49 (73%) domestic, n =32 (48%) international". This fact should be explained in detail in the discussion as in 2003, the government developed a new funding mechanism, which entailed funding tertiary institutions based on their research outputs. Additionally, A national survey of research and development in 2008/2009 recorded gross domestic expenditure on research and development of 1.5 billion Dollars, representing a 13% nominal increase over that of 2007/2008.

5- The study entitled advanced research based on the technology used like fluorescence microscopy, molecular biology, or cell culture work. I don't know why they didn't include advanced Neuro-imaging?

6- In the methods section, there is a need for a flowchart that illustrates the search, data extraction, and curation.

7- Could the authors add a detailed paragraph explaining the tools used for statistical analysis

8- Finally, could the authors write a paragraph in the discussion section explaining how the African Neuroscientists attract funding bodies to invest more in Africa?

Prof. Foad Abd-Allah

Professor of Neurology, and Stroke Medicine, Cairo University, Cairo, Egypt

President of African Academy of Neurology

Reviewer #2 (Remarks to the Author):

The authors report on neuroscience research output in Africa over a period of 20 years and attempt to place their findings in the context of more advanced economies. The manuscript is well written and easy to follow. The approach is commendable in highlighting the output of research in Africa in this fast moving research field. However, I have quite a few concerns. I think the paper is not acceptable in its current form. Major revisions are needed.

1. The title must be changed. "Waking a sleeping giant" does not sound appropriate. In my opinion, this suggests that African countries were once a powerhouse in neuroscience but have since declined.

However, the authors show the opposite – a recent rise in neuroscience output.

2. In the Abstract, the link between the sentence reporting 12,326 publications and the other showing 5,219 is unclear. The reader is left guessing that the latter number came out of further processing of the former.

3. Inconsistencies- several numbers do not match throughout the manuscript.

a. For example, in the Abstract and some other parts of the text, 220 random publications from other countries were used for comparison but 200 is quoted elsewhere.

b. The article selection is said to be from 1996 but in other parts of the manuscript it is stated as 1997. Which is correct?

4. The authors used senior or corresponding authorship to determine if the work was done in Africa. I find this problematic and could skew their findings. This is because, as they say, many African scientists tend to collaborate outside the continent. In such cases where part of the work is done in Africa and another part abroad, it is likely the Africa-led author will be first author (since young scientists are more likely to prefer first authorship), and their (more senior) overseas collaborator as the senior author. However, per the classification system used, such publications will be removed from their analysis since it will be categorised as work done outside Africa.

5. How did the authors define sub-fields and how accurate is this? The info provided is inadequate

6. What are the significance values for the correlation coefficients provided in Fig 1D-G?

7. Why choose the specific periods of 1996 (or 1997) to 2017_ Any motivation?

8. They should show trends in research topics over the period studied

9. There is no info on how data on flight connections were generated

10. The authors discuss in a negative way the apparent lack of model system-based research. However, the major scientific and health issues on the continent have to do with specific (and in some cases peculiar) infectious and non-communicable diseases. Hence, I would expect the primary focus of the scanty funding available is on documenting the public health basis of the disease instead of fancy animal modelling (which would be secondary pending adequate funding). In this context, the authors should discuss research on neuro basis of infectious diseases they found.

11. They discuss extensively research on natural products mainly in West Africa but fail to cover the focus on "other topics" in South and North Africa.

12. As the authors acknowledged, considerable amount of research is published in non-PubMed indexed journals. I would imagine one reason for this is a seeming lack of broad interest of Africa-specific questions to a global audience (and hence fewer citations) that high impact journals often tend to focus on. However, such findings may provide policy-relevant insights into the true neuroscience interests in Africa compared with the relatively few ones covered here. I suggest the authors analyse these non-indexed publications as well.

We thank the two reviewers for their time and very helpful comments, which we have sought to address in full in the below.

Reviewer #1

In the current article, the authors analyzed all of Africa's Neuroscience research output over the past 21 years. Although there are few reports that were published on the same topic, yet the previous estimates addressed global African Neuroscience research, not the African-based neuroscientists. The current study focuses on research output that originally performed in Africa under supervision from African neuroscientists and not the internationally led collaboration. In fact, this effort is genuine and reflects the real metrics of neuroscience research activities in Africa. Furthermore, the authors compared their dataset to findings from well-developed countries in terms of demographic findings, the effect of the economy, and indicators of mobility. From my point of view, this work is valuable however there are some points to be discussed with the authors.

We thank the reviewer for their supportive and insightful comments.

In the abstract line no. 4 the authors mentioned that "In contrast, with an estimated one neuroscientist per million people in Africa, news about neuroscience research from the Global South remains sparse". The accurate and updated Population, Estimated Neurologist Density in Africa is reported in WHO atlas of Neurology 2017 and updated 2020, please try to incorporate in your study.

We thank the reviewer for this suggestion and have now added the points about neurologists to the discussion. For the abstract, we aimed to keep the comment as general as possible, and we think that neuroscientist is a more encompassing term than neurologist, which specifically implies direct human/medical relevance.

In the results section Fig (1). The authors highlighted the relation between neuroscience publications output and research and development (R&D) spending in purchasing power parity dollars (PPP\$). Could the authors explain the term (PPP\$) anywhere in the text as some readers are not oriented by this indicator.

We have now updated the manuscript to include this definition (lines 445-447).

In the results section, the authors divided neuroscience research into nine broad topics according to reference no.18 and according to their research schemes neurodegeneration and injury (n = 2,066, 34%; compared to 22% outside of Africa). The authors must explain why in particular Neurodegenerative research is significantly more productive inside Africa than that led by non-African

Our analysis reflects the relative contribution to diverse research fields within Africa. In absolute numbers, neurodegeneration research remains far more prevalent outside of Africa. Within Africa, we think this particular interest in Neurodegeneration research may in part reflect a particular awareness of African neuroscientists and the public about the high prevalence of central nervous system infections, neurological disorders, and traumatic brain injury (TBI) on the continent. We have updated the manuscript to discuss this point at some depth (lines 304-319).

In terms of funding African neuroscience, the authors mentioned an important fact "only African region where the number of domestic funding mentions exceeded international funder mentions was South Africa (n = 49 (73%) domestic, n =32 (48%) international". This fact should be explained in detail in the discussion as in 2003, the government developed a new funding mechanism, which entailed funding tertiary institutions based on their research outputs. Additionally, A national survey of research and development in 2008/2009 recorded gross domestic expenditure on research and development of 1.5 billion Dollars, representing a 13% nominal increase over that of 2007/2008.

We thank the reviewer for directing our attention to this information. We have updated the manuscript accordingly (lines 279 – 285).

The study entitled advanced research based on the technology used like fluorescence microscopy, molecular biology, or cell culture work. I do not know why they didn't include advanced Neuroimaging?

Yes, this was counted as advanced techniques. We have updated the list accordingly in the methods section.

In the methods section, there is a need for a flowchart that illustrates the search, data extraction, and curation

We have added the requested flowchart (Figure S1).

Could the authors add a detailed paragraph explaining the tools used for statistical analysis?

Thank you for raising this point. The only non-standard statistics used in the manuscript are the so-called generalized additive models (GAMs) shown at the bottom of Figure 1. These are elaborated on in the methods section.

Other more standard statistical analyses are remarked upon in the associated figure legends. For example, Figure3A1 uses a 1-tailed Wilcoxon Rank sum test. Beyond this, data presented are either counts or percentages which are not statistically contrasted against one another

Finally, could the authors write a paragraph in the discussion section explaining how African Neuroscientists can attract funding bodies to invest more in Africa?

Thank you for this very helpful idea. In response, and based on our findings, we think improving domestic funding needs to be prioritized over international funding. For this, we have substantially edited the discussion to suggest general avenues that may be worth exploring (lines 286 – 303).

Reviewer #2

The authors report on neuroscience research output in Africa over a period of 20 years and attempt to place their findings in the context of more advanced economies. The manuscript is well written and easy to follow. The approach is commendable in highlighting the output of research in Africa in this fast-moving research field. However, I have quite a few concerns. I think the paper is not acceptable in its current form. Major revisions are needed.

The title must be changed. “Waking a sleeping giant” does not sound appropriate. In my opinion, this suggests that African countries were once a powerhouse in neuroscience but have since declined. However, the authors show the opposite – a recent rise in neuroscience output

Thank you, we agree. We have changed the title to “20 years of Neuroscience: Unlocking Africa’s potential”

In the Abstract, the link between the sentence reporting 12,326 publications and the other showing 5,219 is unclear. The reader is left guessing that the latter number came out of further processing of the former.

We have clarified this point in the abstract (lines 47-48)

The reviewer raised issues of inconsistencies in the manuscript.

- a. For example, in the Abstract and some other parts of the text, 220 random publications from other countries were used for comparison but 200 is quoted elsewhere.
- b. The article selection is said to be from 1996 but in other parts of the manuscript it is

Thank you for drawing these points to our attention, which have now been rectified as needed.

a. We analyzed 220 papers per non-African country (10 per each of 22 years), however ~20 were discarded per country based on the criteria elaborated in the methods, leaving ~200. However, we do agree that this is confusing, and we have reverted to 220.

b. We apologize for the error. The analysis was conducted on retrieved articles from 1996 – 2017. We have corrected this in the manuscript.

The authors used senior or corresponding authorship to determine if the work was done in Africa. I find this problematic and could skew their findings. This is because, as they say, many African scientists tend to collaborate outside the continent. In such cases where part of the work is done in Africa and another part abroad, it is likely the Africa-led author will be first author (since young scientists are more likely to prefer first authorship), and their (more senior) overseas collaborator as the senior author. However, per the

classification system used, such publications will be removed from their analysis since it will be categorized as work done outside Africa

We agree that using senior/corresponding authorship as the sole criteria to determine eligibility would skew the findings. However, this criterion was only one of several indications used.

In particular, we also noted the study location (based on information from the methods or acknowledgements sections) to evaluate whether the work was done in Africa. In cases where only a small fraction of work was done in Africa, such as plant extract collection, the study was excluded even if the authors were from Africa. When the location of the study was not clear, the study was also excluded. In contrast, studies were included even if the authors did not have an affiliation with an African institution, as long as the actual work was conducted within Africa. We have updated the methods section to provide additional clarification (lines 409-414).

How did the authors define sub-fields and how accurate is this? The info provided is inadequate

The sub-fields were taken 1:1 from the classification provided in [2], which was in turn taken from the SfN "subject themes" from their last conference. The methods section has now been edited to explicitly explain this (lines 433-434).

What are the significance values for the correlation coefficients provided in Fig 1D-G?

This has been added to legend, thank you.

Why choose the specific periods of 1996 (or 1997) to 2017_ Any motivation?

Our general goal was to chart contemporary research outputs, themes, funding, and collaborations in African neuroscience as comprehensively as reasonably possible, so as to highlight areas of strengths and weakness that would be useful for the purpose of making interventions to boost African neuroscience.

Since the Society of Neuroscientists of Africa (SONA) was founded in 1993, we chose to start our analysis from 1996, hoping that from that period, there would have been an increased, perhaps more formally consolidated interest in neuroscience research in Africa.

We decided to stop at 2017 since we began curating papers in early 2018, and we had to define an end point to this analysis in order to not constantly catch up with new papers. Manual screening and curation of the resultant >10,000 papers took us nearly 2 years, and we believe that the resultant dataset, which is freely available already now, can provide a rich resource for understanding publication trends over a substantial timeframe.

Finally, some published papers, especially from some African journals, take years to become available online. Thus, by choosing to stop at 2017, we likely by now included all articles published up to 2017.

They should show trends in research topics over the period studied.

We thank the reviewer for this suggestion. In fact, we had done this analysis, and found that there was very little change in this distribution over the past 20 years:

Accordingly, we decided to leave this panel out to make room for different analyses, and to instead remark on this finding in writing (lines 128-130).

There is no info on how data on flight connections were generated.

The data were obtained from www.OpenFlights.org, as noted in the Methods section and the corresponding figure legend.

OpenFlights is an open-source tool that enables anyone to map flights around the world and statistically analyse the density of flights between different destinations. The underlying dataset for OpenFlights.org is publicly available at: <https://github.com/jpatokal/openflights/>. To further clarify this point, we added this direct link to the methods section (lines 440-442)

The authors discuss in a negative way the apparent lack of model system-based research. However, the major scientific and health issues on the continent have to do with specific (and in some cases peculiar) infectious and non-communicable diseases. Hence, I would expect the primary focus of the scanty funding available is on documenting the public health basis of the disease instead of fancy animal modelling (which would be secondary pending adequate funding). In this context, the authors should discuss research on neurobasis of infectious diseases they found.

We thank the reviewer for their comment. Our work's focus was to provide a broad overview of neuroscience research in Africa that would guide future strategies to boost this sector. Beyond direct study of disease, there are many basic neuroscientists on the continent, some of which are

working to understand the fundamental basis of brain function. We believe it is therefore crucial to also analyze the type of work they do, also in view of model systems. We have already seen the advantages of this through the activities of many organizations working to introduce cheap, yet powerful model systems for neuroscience research in Africa. Moreover, much of the rest of the world is investing heavily in these model systems, so to better synergize internationally, and to compare results, we believe that the promotion of model systems, and all the advantages that they bring (especially genetic access), will be critical in the long run.

However, we agree that it is equally important to discuss data on more applied research around central nervous system infections and disorders. Accordingly, we have extended the discussion to highlight some of the publication trends regarding CNS infections and disorders, literature around this and how that can help attract funding for the discipline (lines 304 – 319).

They discuss extensively research on natural products mainly in West Africa but fail to cover the focus on “other topics” in South and North Africa.

We thank the reviewer for this comment. We fully agree that the particular strengths of neuroscience research in Africa are diverse across the regions, and in general we have sought to avoid a special regional focus on research themes. This is for example attempted by Figure 1, which revealed that some major topics of investigation across essentially all of Africa are, despite inevitable local variations, surprisingly homogeneous, and consistently centered around neurodegeneration and injury.

Regarding specifically medicinal plant usage in neuroscience: judging from themes presented at recent SONA conferences which may arguably be taken as approximately representative for African neuroscience, this is consistently a topic of particular interest. This may be linked to its particular history in Africa [3]. However, to our surprise here we moreover found that research on this particular topic is heavily skewed to a small number of African countries that are located near the equator. However, some of these countries are disproportionately populous (e.g. Nigeria, Cameroon), which might be linked to the strong representation of this field at conferences.

Importantly, all collected data is available free to download, and there are many aspects which have not been analyzed here. Rather, we see our work as a resource for others to use and draw their own conclusions, with the manuscript presented providing a hopefully useful overview of some of the major trends. Doubtless, there are many additional insights to be gained from further analysis, and we will be excited to see what analyses others may produce going forward.

As the authors acknowledged, considerable amount of research is published in non-PubMed indexed journals. I would imagine one reason for this is a seeming lack of broad interest of Africa-specific questions to a global audience (and hence fewer citations) that high impact journals often tend to focus on. However, such findings may provide policy-relevant insights into the true neuroscience interests in Africa compared with the relatively few ones covered here. I suggest the authors analyze these non-indexed publications as well.

We thank the reviewer for their suggestions. We fully agree that ideally all published and peer-reviewed research would be included, including those articles not indexed on PubMed.

However, we think that PubMed-indexing is the best tool available today to provide a mostly unbiased overview of the most visible neuroscience research done in Africa. It is undeniably a key international standard, and the papers found here are by far the most likely to be seen, read, and cited.

Moreover, we worry that analyzing non-PubMed indexed neuroscience research from Africa may pose the risk of producing more problems than solutions:

- 1. Unlike PubMed, we are not aware of a single, unified and standardized source to ensure homogenous retrieval of articles from non-indexed African journals.*
- 2. Individually searching selected African Journal websites would also be limiting because this strategy would be likely to miss low-profile journals, providing additional bias. Moreover, the search engines of some African journal websites are poorly developed (or non-existent), meaning that it may prove impossible to find some of the articles in the first place. Further, some African journals are not available online, and those that are, are not necessarily accessible without specific memberships (which would be difficult to obtain).*
- 3. Unfortunately, a fraction of African research articles is published in non-indexed predatory journals. It is likely that some of these articles would not have passed peer-review, which was an exclusion criterion for our analysis.*
- 4. Much of our analysis uses citation counts, which are often either unavailable for non-indexed journals, or compiled by different metrics than those used for indexed publications. Accordingly, combining these datasets would likely be very problematic.*

References

- [1] Russell, V.A., 2017. Notes on the recent history of neuroscience in Africa. *Frontiers in neuroanatomy*, 11, p.96.
- [2] Carandini, M., 2019. Charting the Structure of Neuroscience. *Neuron*, 102(4), pp.732-734.
- [3] Kuete, V. ed., 2013. *Medicinal plant research in Africa: Pharmacology and chemistry*. Newnes.

REVIEWER COMMENTS

Reviewer #1 (Remarks to the Author):

Thanks to the authors, almost all concerns have been addressed and answered appropriately.

Reviewer #2 (Remarks to the Author):

I thank the authors for their thorough revision of the manuscript and rebuttal to the reviewer comments. This article and the open access resource they provide will be highly impactful for science funding policy in Africa. I have no further comments.

Thomas K. Karikari
University of Gothenburg, Sweden/University of Pittsburgh, USA

Reviewer #3 (Remarks to the Author):

This paper attempts to describe the extent to which there is variation across African countries in the production of neuroscience publications. This kind of work is important but there are several issues with the manuscript that I see as important to address before moving forward.

I outline my major comments below, followed by some minor comments.

1. It is not clear from the introduction what exactly the research question is. The paper is set up such that I am expecting a comparison of research carried out in Africa versus research that has African authors, but is conducted mostly outside the continent. However, once I got to the results section, it appears that the paper is presenting descriptive statistics of the features of research carried out in Africa. This is a worthwhile pursuit, particularly if more is done with regards to the variation over time and across countries, but the introduction is somewhat misleading.
2. Given the focus on the importance of understanding more about Africa-led research in the introduction section, I would like a clearer definition of 'Africa-led' earlier on in the paper. There are several ways that this could be defined: research could be 'led' by researchers in an African country, in that they provide the funding or intellectual input; the samples could be collected in an African country; or the analysis could be carried out in an Africa country.
3. Relatedly, I would like to know more about why this is important. It isn't clear to me why a global division of labor is a problem. Or why we need to know more about it, particularly as it is defined by the

authors.

4. The methods used to carry out the correlational analysis (results presented in Figure 1: D E F G) is very unclear. For example, do you hold year constant? And include country fixed effects? This would need to be clarified in both the main text, the figure notes, and the methods section. Also, the language needs to be such that makes it clear that this is not a claim of causality. But actually, I would recommend the authors remove this entirely – it is not central to their story, and so I recommend the authors should stick to the descriptive facts.

Minor comments:

1. The abstract should be shortened.
2. Grammar and language should be tightened. There are quite a few mistakes and confusing sentences.
3. The time-frame of analysis is not clear (you say since 1996 – but when did the data collection stop)?
4. It is unclear in the first part of the results section if the authors are talking about overall publications with an author address from an African country? Or whether this refers to the set of 5,219 publications?
5. I feel like there are some unfounded statements like on line 120 “one might expect this trend to continue going forward”.
6. Data extraction procedure: is the search term using African country names in the affiliation field? Or in the text of the documents? This wasn't clear.

Reviewer #4 (Remarks to the Author):

In “20 years of Neuroscience: Unlocking Africa’s potential” the authors conduct a systematic review of the neuroscience literature produced in Africa over the past two decades. The authors assess the national characteristics associated with research output, describe regional and international differences in topical focus, measure links between team type/location and papers’ impact, and investigate domestic and international funding sources. The paper is relevant and well-written, and the authors responded comprehensively to a previous round of reviews. At this point, I have a few remaining substantive recommendations, and several minor comments.

General: If possible, I would be very interested to see figures and/or discussion around changes in funding sources and intra/inter-regional collaboration over time. Two decades is a long time, and aggregating over the full 1996-2017 window could obscure some relevant temporal information. Quantifications/visualizations of temporal trends could provide additional insight into current obstacles and opportunities for the field, especially when paired with the graphics and discussions already presented in the manuscript.

Line 117: In the text the authors describe exponential increases in output across all regions, but the

stacked figure (Fig 1B) makes it hard to see increases for any region other than “North”. Perhaps the authors could include an additional figure where each region has its own line showing “% increase in publications relative to 1996” on the y-axis. This would make it easier to assess and compare temporal trends across regions. Such a graph could go in the supplement if it cannot fit in Fig 1.

Line 130: It makes sense that the figure showing research topics over time (provided in the first response to reviewers) will not be included in the main text. However, it would be useful to include this figure in the supplement. It is likely to be of interest to readers.

Line 194: Why does the funding analysis only include papers published in journals with impact factors greater than or equal to 5? The authors should add a sentence in the methods describing the rationale behind this subset.

Figure 3: It is pretty difficult to compare panels C/D to panels E/F given the density of lines. Perhaps you could link panel B with C/D, and then create a graph similar to panel B that visualizes inter-/intra-region flight connectivity. For a given region in Africa, region A, cell shading could potentially be calculated as (# of flights between region A and region B) / (# of flights into or out of region A).

Line 233: Using “specifically” here makes it sound like these three are the only techniques considered to be advanced techniques. Replacing “specifically” with “for example” might be clearer, since these three appear to be only a subset of the many advanced techniques that the authors list in the methods.

Line 265: The sentence starting with “Similarly” is hard to follow. Perhaps the section about neurologists could be split off into its own sentence.

Line 391 + Fig S1: Additional detail on article exclusion would be helpful. For example, it would be helpful to know how many of the excluded papers were removed because they were A) review articles, B) duplicates, C) irrelevant, D) not carried out in the relevant location. From the methods it seems that criterion D was the primary driver behind removing 58% of the candidate African papers, but it is unclear which of the above criteria contributed to removing ~20% of non-African papers. Ideally this information could be included in the supplementary flow diagram.

Line 405: In general, it is important to include more detail on the raters who conducted the data curation. Specifically, what training did these curators have, and how many curators were responsible for manually rating papers’ location, techniques, and topics? Were any papers rated by multiple curators in order to assess agreement and reliability of ratings, or was it done through a consensus process?

Reviewer #1

Thanks to the authors, almost all concerns have been addressed and answered appropriately.

Thank you again for taking the time for carefully considering our work, and for your support in improving it in view of getting it published.

Reviewer #2

I thank the authors for their thorough revision of the manuscript and rebuttal to the reviewer comments. This article and the open access resource they provide will be highly impactful for science funding policy in Africa. I have no further comments.

Thank you again for taking the time for carefully considering our work, and for your support in improving it in view of getting it published.

Reviewer #3

This paper attempts to describe the extent to which there is variation across African countries in the production of neuroscience publications. This kind of work is important but there are several issues with the manuscript that I see as important to address before moving forward.

I outline my major comments below, followed by some minor comments.

1. It is not clear from the introduction what exactly the research question is. The paper is set up such that I am expecting a comparison of research carried out in Africa versus research that has African authors, but is conducted mostly outside the continent. However, once I got to the results section, it appears that the paper is presenting descriptive statistics of the features of research carried out in Africa. This is a worthwhile pursuit, particularly if more is done with regards to the variation over time and across countries, but the introduction is somewhat misleading.

Following the reviewers' advice, we have now added an explicit set of research questions to the end of the introduction section:

"Specifically, we asked: How many publications came out of each African country, and how were different sub-disciplines represented (Fig. 1), in which journals are they published and how many citations did they attract (Fig. 2), what were the major trends of international collaborations (Fig. 3), how was the work funded (Fig. 4), and what experimental techniques were used (Fig. 5)?"

2. Given the focus on the importance of understanding more about Africa-led research in the introduction section, I would like a clearer definition of 'Africa-led' earlier on in the paper. There are several ways that this could be defined: research could be 'led' by researchers in an African country, in that they provide the funding or intellectual input; the samples could be collected in an African country; or the analysis could be carried out in an Africa country.

We have now added the following definition to the introduction:

“Here, we use “African led” to mean publications with clear evidence that that the bulk of the intellectual input and experimental work was carried out by researchers who are primarily based at African institutions (Methods).”

3. Relatedly, I would like to know more about why this is important. It isn't clear to me why a global division of labor is a problem. Or why we need to know more about it, particularly as it is defined by the authors.

A major motivation for our survey was to identify the real contribution and impact of African (neuro)science research on the global stage, and to systematically understand what research infrastructure is available across Africa's research centres. Knowing these types of parameters will be instrumental to support African research in a meaningful and sustainable manner, by funders, governments, and by the academics themselves.

Division of labour can be useful, but not when there is so much disparity between researchers in the Global North and South - it makes it very difficult for any such division to yield equal reward for all involved. For example, from experience, African researchers are rarely enabled to take a leading role in multinational collaborations because they usually do not have the necessary infrastructure in place to contribute on an equal footing.

Our survey systematically showcases, using data, systematic impasses across Africa's neuroscience labs: perhaps most importantly, a near-complete absence of even basic infrastructure in all but a handful of internationally sponsored (and usually also internationally led) institutes. This has dire consequences not only for research, but for the whole education sector and society. Many of the best African researchers move abroad, and those that stay must conduct research with poor infrastructure support, alongside a typically high teaching load. This in turn damages teaching quality and hurts graduates from Africa's universities who will eventually form the next generation of the continent's leaders in research, business, and politics.

These types of deeply interconnected issues can only be meaningfully broken by sustained and large-scale shifts in policy. However, impactful policy needs accurate data, and with our work we hope to have contributed to this goal in a small manner.

4. The methods used to carry out the correlational analysis (results presented in Figure 1: D E F G) is very unclear. For example, do you hold year constant? And include country fixed effects? This would need to be clarified in both the main text, the figure notes, and the methods section. Also, the language needs to be such that makes it clear that this is not a claim of causality. But actually, I would recommend the authors remove this entirely – it is not central to their story, and so I recommend the authors should stick to the descriptive facts.

We have followed the reviewer's suggestion to remove the four panels.

Minor comments:

1. The abstract should be shortened.

The abstract has now been cut to 150 words to conform with formatting requirements at Nature Communications.

2. Grammar and language should be tightened. There are quite a few mistakes and confusing sentences.

We have carefully gone through the MS again and amended any statements that we thought might be confusing.

3. The time-frame of analysis is not clear (you say since 1996 – but when did the data collection stop)?

In 2017, as now noted in the introduction and the title to figure 1, and further justified in the Methods.

4. It is unclear in the first part of the results section if the authors are talking about overall publications with an author address from an African country? Or whether this refers to the set of 5,219 publications?

This is for the 5,219 publications that passed our inclusion criteria, as now explicitly noted in the first sentence of the results section.

5. I feel like there are some unfounded statements like on line 120 “one might expect this trend to continue going forward”.

Thank you, this is now rephrased to simply point at the recent upwards trajectory.

6. Data extraction procedure: is the search term using African country names in the affiliation field? Or in the text of the documents? This wasn't clear.

It is using them in both the text and in the affiliation field. From here, we manually excluded those papers that only mentioned specific African countries with the research having been done elsewhere. This is now noted in the Methods.

Reviewer #4

In “20 years of Neuroscience: Unlocking Africa’s potential” the authors conduct a systematic review of the neuroscience literature produced in Africa over the past two decades. The authors assess the national characteristics associated with research output, describe regional and international differences in topical focus, measure links between team type/location and papers’ impact, and investigate domestic and international funding sources. The paper is relevant and well-written, and the authors responded comprehensively to a previous round of reviews.

Thank you.

At this point, I have a few remaining substantive recommendations, and several minor comments.

General: If possible, I would be very interested to see figures and/or discussion around changes in funding sources and intra/inter-regional collaboration over time. Two decades is a long time, and aggregating over the full 1996-2017 window could obscure some relevant temporal information. Quantifications/visualizations of temporal trends could provide

additional insight into current obstacles and opportunities for the field, especially when paired with the graphics and discussions already presented in the manuscript.

We thank the reviewer for this suggestion. We had previously considered something similar, however the problem is that for breaking these metrics down by year the absolute numbers become quite small. To illustrate this problem, we now quantified the fraction of domestic funding in IF \geq 5 publications by region and time, binned into 6-year instances. Even with this rather coarse time-resolution several regions had fewer than ten included publications that also had a funding declaration, which renders conclusions drawn rather fragile. For example, by fraction (top graph) it is tempting to suggest that Central and West Africa used to have some domestic funding in the late 1990s which ceased in the early 2000s before recovering in more recent years. However, the absence of domestic funding declarations in the two middle bins is based on only 2-7 publications per instance (bottom graph), which does not provide sufficient statistical power to draw solid conclusions. Accordingly, we have opted to not include analyses of this type.

Regarding the second point on collaborations, in this case the absolute numbers are much larger, so breaking them down by year is possible. To this end, we now computed the collaboration matrix (Fig. 3B) for three instances in time:

This revealed that by and large, and despite small local variations, collaboration trends were approximately stable overall for the surveyed time-period. However, there was a general increase in the fraction of publications that had international collaborations (here seen in the general “darkening” of the matrices from left to right). The above panels are now included in the Supplement (SFig. 2C), and noted in the results section:

“Generally, fractions of international collaborations increased over the study period, with approximately preserved geographical patterns (Fig. S2C).”

Line 117: In the text the authors describe exponential increases in output across all regions, but the stacked figure (Fig 1B) makes it hard to see increases for any region other than “North”. Perhaps the authors could include an additional figure where each region has its own line showing “% increase in publications relative to 1996” on the y-axis. This would make it easier to assess and compare temporal trends across regions. Such a graph could go in the supplement if it cannot fit in Fig 1.

This has been added to the supplement (Fig S2).

Line 130: It makes sense that the figure showing research topics over time (provided in the first response to reviewers) will not be included in the main text. However, it would be useful to include this figure in the supplement. It is likely to be of interest to readers.

Thank you, these have now been added to the supplement as suggested.

Line 194: Why does the funding analysis only include papers published in journals with impact factors greater than or equal to 5? The authors should add a sentence in the methods describing the rationale behind this subset.

We have added the requested information to the Methods:

“We used only this subset of publication because this task was unusually time-consuming in view of incomplete standardisation across publishing outlets and the large diversity of funders worldwide.”

Figure 3: It is pretty difficult to compare panels C/D to panels E/F given the density of lines. Perhaps you could link panel B with C/D, and then create a graph similar to panel B that visualizes inter-/intra-region flight connectivity. For a given region in Africa, region A, cell shading could potentially be calculated as (# of flights between region A and region B) / (# of flights into or out of region A).

We thank the reviewer for this suggestion, which we agree seems like a worthwhile extension. However, when aiming to implement this idea, we encountered a number of issues which limit our confidence that an analysis of this type would be more useful than misleading.

One major limitation is that the flight data has information on the existence of a route, but not on the frequency at which it is serviced, nor the type of airplane or any information on the numbers of passengers. These types of information would however be critical to meaningfully compute a quantitative matrix that could be compared with the numerically much more well-founded matrix on collaborations. Another, possibly more minor problem is that Africa has a rather dominant “hub” structure which would distort quantitative region-wise measures of travel (illustrated here based on available data – note for example the dominance of Johannesburg and Nairobi intra-African

connections (bottom, red) or the dominance of North African airports such as Cairo, Algiers, Tunis, Marrakesh for linking to Europe (top map, back).

Accordingly, we think it is less misleading to simply refer to the flights data as an interesting parallel to collaboration networks, and as an illustrative example of how difficult it can be to directly travel between two African countries without passing through e.g. Europe or the Middle East. This makes it logistically challenging to establish and maintain meaningful intra-African collaborations over long distances, unless they happen to align with a dominant flight-corridor.

Line 233: Using “specifically” here makes it sound like these three are the only techniques considered to be advanced techniques. Replacing “specifically” with “for example” might be clearer, since these three appear to be only a subset of the many advanced techniques that the authors list in the methods.

Thank you, we have now followed this suggestion.

Line 265: The sentence starting with “Similarly” is hard to follow. Perhaps the section about neurologists could be split off into its own sentence.

Thank you, we have now rephrased this section along the suggested lines.

Line 391 + Fig S1: Additional detail on article exclusion would be helpful. For example, it would be helpful to know how many of the excluded papers were removed because they were A) review articles, B) duplicates, C) irrelevant, D) not carried out in the relevant location. From the methods it seems that criterion D was the primary driver behind removing 58% of the candidate African papers, but it is unclear which of the above criteria contributed to removing ~20% of non-African papers. Ideally this information could be included in the supplementary flow diagram.

10% of exclusion were duplicates (now noted). However, beyond this we did not track the specific fractions of criteria for exclusion, which were often also overlapping.

Line 405: In general, it is important to include more detail on the raters who conducted the data curation. Specifically, what training did these curators have, and how many curators were responsible for manually rating papers’ location, techniques, and topics? Were any papers rated by multiple curators in order to assess agreement and reliability of ratings, or was it done through a consensus process?

All raters were trained neuroscientists, with experience ranging from MSc level to faculty. Rating practise was aligned (1) by the lead author training all raters personally, (2) each paper was independently rated twice, by two different members of the team followed by (3) manual checking and as necessary adjusting by the first author for any inconsistencies between each paper’s two ratings. This is now included in the Methods.

REVIEWERS' COMMENTS

Reviewer #3 (Remarks to the Author):

Thank you for addressing the comments in the previous round of reviews. I think the paper is much strengthened, and I still think that it is a valuable contribution to the literature and of interest to readers of this journal.

However, there are still some outstanding issues that I suggest should be addressed prior to publication:

- The manuscript still contains a large number of typos and grammatical errors.

- In addition there are a large number of sentences that require further clarification. For example, in the first paragraph (line 62) – what do you mean the publications have been rising? But then you say 70% of them have no African author. Are you talking about publications using African data? About Africa? Can you be more specific? That is just one example.

- Thank you for indicating your definition of African-led early on (line 68). I think this helps, but I think you need to go even further here given that this is critical for your study. Is this a new measure you are creating? Has it been used in the literature before?

- Similarly – around line 86 I think you need to say what your definition of African neuroscience publications is.

Reviewer #4 (Remarks to the Author):

I thank the authors for their careful consideration of the reviewers' comments. They have directly incorporated most of the provided suggestions, and they have responded thoughtfully and thoroughly to the rest. At this point I have no further comments.

REVIEWERS' COMMENTS

Reviewer #3 (Remarks to the Author):

Thank you for addressing the comments in the previous round of reviews. I think the paper is much strengthened, and I still think that it is a valuable contribution to the literature and of interest to readers of this journal.

However, there are still some outstanding issues that I suggest should be addressed prior to publication:

- The manuscript still contains a large number of typos and grammatical errors.

We have gone through carefully and corrected any that we could find.

- In addition there are a large number of sentences that require further clarification. For example, in the first paragraph (line 62) – what do you mean the publications have been rising? But then you say 70% of them have no African author. Are you talking about publications using African data? About Africa? Can you be more specific? That is just one example.

The 70% pertains to a number from a previous publication, where the authors report that 70% of papers from Sub Saharan Africa had at least one non-African co-author. This is now further clarified

- Thank you for indicating your definition of African-led early on (line 68). I think this helps, but I think you need to go even further here given that this is critical for your study. Is this a new measure you are creating? Has it been used in the literature before?

We have previously used this measure in a similar study that specifically assesses neuroscience publications from Nigeria (Maina et al. EJN). This is now further referenced, and in addition we now further detail how the metric was applied.

- Similarly – around line 86 I think you need to say what your definition of African neuroscience publications is.

Since we have now explicitly defined this twenty lines prior (see previous comment), we here simply refer to this definition.